# SAFE REINFORCEMENT LEARNING FROM PIXELS USING A STOCHASTIC LATENT REPRESENTATION

**Yannick Hogewind, Thiago D. Simão, Tal Kachman & Nils Jansen**
Radboud University, Nijmegen
{yannick.hogewind,thiago.simao,nils.jansen}@ru.nl
tal.kachman@donders.ru.nl

## ABSTRACT

We address the problem of safe reinforcement learning from pixel observations. Inherent challenges in such settings are (1) a trade-off between reward optimization and adhering to safety constraints, (2) partial observability, and (3) high-dimensional observations. We formalize the problem in a constrained, partially observable Markov decision process framework, where an agent obtains distinct reward and safety signals. To address the curse of dimensionality, we employ a novel safety critic using the stochastic latent actor-critic (SLAC) approach. The latent variable model predicts rewards and safety violations, and we use the safety critic to train safe policies. Using well-known benchmark environments, we demonstrate competitive performance over existing approaches regarding computational requirements, final reward return, and satisfying the safety constraints.

## 1 INTRODUCTION

As reinforcement learning (RL) algorithms are increasingly applied in the real-world (Mnih et al., 2015; Jumper et al., 2021; Fu et al., 2021), their safety becomes ever more important with the increase of both model complexity, and uncertainty. Considerable effort has been made to increase the safety of RL (Liu et al., 2021). However, major challenges remain that prevent the deployment of RL in the real-world (Dulac-Arnold et al., 2021). Most approaches to safe RL are limited to fully observable settings, neglecting issues such as noisy or imprecise sensors. Moreover, realistic environments exhibit high-dimensional observation spaces and are largely out of reach for the state-of-the-art. In this work, we present an effective safe RL approach that handles partial observability with high-dimensional observation spaces in the form of pixel observations.

In tandem with prior work, we formalize the safety requirements using a constrained Markov decision process (CMDP; Altman, 1999). The objective is to learn a policy that maximizes a reward while constraining the expected return of a scalar cost signal below a certain value (Achiam et al., 2017). According to the reward hypothesis, it could be possible to encode safety requirements directly in the reward signal. However, as argued by Ray et al. (2019), safe RL based only on a scalar reward carries the issue of designing a suitable reward function. In particular, balancing the trade-off between reward optimization and safety within a single reward is a difficult problem. Moreover, over-engineering rewards to complex safety requirements runs the risk of triggering negative side effects that surface after integration into broader system operation (Abbeel & Ng, 2005; Amodei et al., 2016). Constrained RL addresses this issue via a clear separation of reward and safety.

Reinforcement learning from pixels typically suffers from sample inefficiency, as it requires many interactions with the environment. In the case of safe RL, improving the sample efficiency is especially crucial as each interaction with the environment, before the agent reaches a safe policy, has an opportunity to cause harm (Zanger et al., 2021). Moreover, to ensure safety, there is an incentive to act pessimistic with regard to the cost (As et al., 2022). This conservative assessment of safety, in turn, may yield a lower reward performance than is possible within the safety constraints.

**Our contribution.** We propose Safe SLAC, an extension of the Stochastic Latent Actor Critic approach (SLAC; Lee et al., 2020) to problems with safety constraints. SLAC learns a stochastic latent variable model of the environment dynamics, to address the fact that optimal policies in partially ob-

servable settings must estimate the underlying state of the environment from the observations. The model predicts the next observation, the next latent state, and the reward based on the current observation and current latent state. This latent state inferred by the model is then used to provide the input for an actor-critic approach (Konda & Tsitsiklis, 1999). This algorithm involves learning a critic function that estimates the utility of taking a certain action in the environment, which serves as a supervision signal for the policy, also named the actor. The SLAC method has excellent sample efficiency in the safety-agnostic partially observable setting, which renders it a promising candidate to adapt to high-dimensional settings with safety constraints. At its core, SLAC is an actor-critic approach, carrying the potential for a natural extension to safety with a safety critic. We extend SLAC in three ways to create our safe RL approach under partial observability (Safe SLAC): (1) the latent variable model also predicts cost violations, (2) we learn a safety critic that predicts the discounted cost return, and (3) we modify the policy training procedure to optimize a safety-constrained objective by use of a Lagrangian relaxation, solved using dual gradient descent on the primary objective and a Lagrange multiplier to overcome the inherent difficulty of constrained optimization.

We evaluate Safe SLAC using a set of benchmark environments introduced by Ray et al. (2019). The empirical evaluation shows competitive results compared with complex state-of-the-art approaches.

## 2 RELATED WORK

Established baseline algorithms for safe reinforcement learning in the fully observable setting include constrained policy optimization (CPO; Achiam et al., 2017), as well as trust region policy optimization (TRPO)-Lagrangian (Ray et al., 2019), a cost-constrained variant of the existing trust region policy optimization (TRPO; Schulman et al., 2015). While TRPO-Lagrangian uses an adaptive Lagrange multiplier to solve the constrained problem with primal-dual optimization, CPO solves the problem of constraint satisfaction analytically during the policy update.

The method closest related to ours is Lagrangian model-based agent (LAMBDA; As et al., 2022), which also addresses the problem of learning a safe policy from pixel observations under high partial observability. LAMBDA uses the partially stochastic dynamics model introduced by (Hafner et al., 2019). The authors take a Bayesian approach on the dynamics model, sampling from the posterior over parameters to obtain different instantiations of this model. For each instantiation, simulated trajectories are sampled. Then, the *worst cost return* and *best reward return* are used to train critic functions that provide a gradient to the policy. LAMBDA shows competitive performance with baseline algorithms, however, there are two major trade-offs. First, by taking a pessimistic approach, the learned policy attains a lower cost return than the allowed cost budget. A less pessimistic approach that uses the entirety of the allowed cost budget may yield a constraint-satisfying policy with a higher reward return. Second, the LAMBDA training procedure involves generating many samples from their variable model to estimate the optimistic/pessimistic temporal difference updates.

While the reinforcement learning literature has numerous safety perspectives (García & Fernández, 2015; Pecka & Svoboda, 2014), we focus on constraining the behavior of the agent on expectation. A method called shielding ensures safety already during training, using temporal logic specifications for safety (Alshiekh et al., 2018; Jansen et al., 2020). Such methods, however, require extensive prior knowledge in the form of a (partial) model of the environment (Carr et al., 2023).

## 3 CONSTRAINED PARTIALLY OBSERVABLE MARKOV DECISION PROCESSES

In reinforcement learning, an agent learns to sequentially interact with an environment to maximize some signal of utility. This problem setting is typically modeled as a Markov decision process (MDP; Sutton & Barto, 2018), in which the environment is composed of a set of states $\mathcal{S}$ and a set of actions $\mathcal{A}$. At each timestep $t$, the agent receives the current environment state $s_t \in S$ and executes an action $a_t \in \mathcal{A}$, according to the policy $\pi$: $a_t \sim \pi(a_t \,|\, s_t)$. This action results in a new state according to the transition dynamics $s_{t+1} \sim p(s_{t+1} \,|\, s_t, a_t)$ and a scalar reward signal $r_t = r(s_t, a_t) \in \mathbb{R}$, where $r$ is the reward function. The goal is for the agent to learn an optimal policy $\pi^\star$ such that the expectation of discounted, accumulated reward in the environment under that policy is maximized, i.e. $\pi^\star = \arg\max_\pi \mathbb{E}\left[\sum_t \gamma^t r_t\right]$ with $\gamma \in [0, 1)$. We use $\rho_\pi$ to denote the distribution over trajectories induced in the environment by a policy $\pi$.

In a partially observable Markov decision process (POMDP; Kaelbling et al., 1998), the agent cannot observe the true state $s_t$ of the MDP and instead receives some observation $x_t \in \mathcal{X}$ that provides partial information about the state, sampled from the observation function $x_t \sim X(s_t)$. In this setting, learning a policy $\pi$ is more difficult than in MDPs since the true state of the underlying MDP is not known and must be inferred from sequences of observations to allow the policy to be optimal. Therefore, the optimal policy is a function of a history of observations and actions $\pi(a_t \mid x_{1:t}, a_{1:t-1})$. In practice, representing such policy can be infeasible, so the policy is often conditioned on a compact representation of the history.

While the reward hypothesis states that desired agent behavior can plausibly be represented in a single scalar reward signal, in practice it can be difficult to define a reward function that balances different objectives (Vamplew et al., 2022; Roy et al., 2022). The same is true for safety: as argued by Ray et al. (2019), a useful definition of safety is a constraint on the behavior, stating the problem as a constrained MDPs (CMDP; Altman, 1999), with analogous definitions for constrained POMDPs (CPOMDP; Isom et al., 2008; Lee et al., 2018; Walraven & Spaan, 2018). A scalar cost variable $c_t \in \mathbb{R}$ at each time step of the C(PO)MDP, according to a cost function $c_t = c(s_t, a_t)$, serves as a measure of safety violation. The objective of the reinforcement learning problem then changes to constrain the accumulated cost under a given safety threshold $d \in \mathbb{R}$:

$$\pi^* = \arg\max_{\pi} \mathbb{E}\left[\sum_t \gamma^t r_t\right] \quad \text{s.t.} \quad \mathbb{E}\left[\sum_t c_t\right] < d. \tag{1}$$

Next, we review RL algorithms for POMDPs and CMDPs, laying the foundations for the safe RL algorithm we propose in Section 5 dedicated to CPOMDPs with unknown environment dynamics.

## 4 STOCHASTIC LATENT ACTOR-CRITIC

Soft actor-critic (SAC; Haarnoja et al., 2018a) is an RL approach based on the maximum entropy framework. Besides the traditional reinforcement learning objective of maximizing the reward, it also aims to maximize the entropy of the policy. The sampling of states that allow high-entropy action distributions is maximized. This approach can be robust to disturbances in the dynamics of the environment (Eysenbach & Levine, 2022). In practice, it can be challenging to determine a weight for the entropy term in the objective a priori, so instead, the entropy is constrained to a minimum value (Haarnoja et al., 2018b). A stochastic policy is particularly interesting to our work since a deterministic policy might be suboptimal in the CPOMDP setting (Kim et al., 2011).

Stochastic latent actor-critic (SLAC; Lee et al., 2020) is an RL algorithm that addresses partial observability and high-dimensional observations by combining a probabilistic sequential latent variable model with an actor-critic approach. The sequential latent variable model aims to infer the true state of the environment by considering the environment as having an unseen true latent state $\mathbf{z}_t$ and latent dynamics $p(\mathbf{z}_{t+1} \mid \mathbf{z}_t, \mathbf{a}_t)$. This model generates observations $\mathbf{x}_t \sim p(\mathbf{x}_t \mid \mathbf{z}_t)$ and rewards $\mathbf{r}_t \sim p(\mathbf{r}_t \mid \mathbf{z}_t, \mathbf{a}_t, \mathbf{z}_{t+1})$. By approximating these latent dynamics in the latent variable model using approximate variational inference, the learned model can infer the latent state from previous observations and actions as $q(\mathbf{z}_t \mid \mathbf{x}_{0:t}, \mathbf{a}_{0:t})$. Consequently, SLAC can be viewed as an adaption of SAC to POMDPs, using the inferred latent state $\mathbf{z}$ as input for the critic in SAC.

Equation 2 describes the latent variable model $M$, which is parameterized by $\psi$. This model infers a low-dimensional latent state representation $\mathbf{z}$ from a history of high-dimensional observations $\mathbf{x}$ and actions $\mathbf{a}$ gathered through interaction with the environment. This model is trained by sampling trajectories from the environment and inferring a latent state from each trajectory according to the distributions given in Equation 3. The model uses this inferred latent state to predict the next observation and reward according to the distributions in Equation 3.

$$
\begin{aligned}
\mathbf{z}_1 &\sim p(\mathbf{z}_1). \\
\mathbf{z}_{t+1} &\sim p_{\psi^z}(\mathbf{z}_{t+1} \mid \mathbf{z}_t, \mathbf{a}_t). \\
\mathbf{x}_t &\sim p_{\psi^x}(\mathbf{x}_t \mid \mathbf{z}_t). \\
\mathbf{r}_t &\sim p_{\psi^r}(\mathbf{r}_t \mid \mathbf{z}_t, \mathbf{a}_t, \mathbf{z}_{t+1}).
\end{aligned}
\tag{2}
\qquad
\begin{aligned}
\mathbf{z}_1 &\sim q_{\psi^0}(\mathbf{z}_1 \mid \mathbf{x}_1). \\
\mathbf{z}_{t+1} &\sim q_{\psi^z}(\mathbf{z}_{t+1} \mid \mathbf{x}_{t+1}, \mathbf{z}_t, \mathbf{a}_t).
\end{aligned}
\tag{3}
$$

Hereinafter, we use $M$ to refer to the functions of the model ($p_{\psi^z}, p_{\psi^x}, p_{\psi^r}, q_{\psi^0}, q_{\psi^z}$), and use $\psi$ without the superscript to simplify the notation. Furthermore, following Lee et al. (2020), in our

implementation we decompose the latent variable $z$ in two parts (see Appendix A for more details). The parameters of the model $\psi$ are trained to optimize the objective in Equation 4, in which $D_{KL}$ is the Kullback-Leibler divergence, an asymmetric measure of the difference between two distributions commonly used in variational inference (Kingma & Welling, 2014):

$$J_M(\psi) = \mathop{\mathbb{E}}_{\mathbf{z}_{1:\tau+1} \sim q_\psi} \left[ \sum_{t=0}^{\tau} \begin{array}{l} -\log p_\psi(\mathbf{x}_{t+1} \mid \mathbf{z}_{t+1}) - \log p_\psi(\mathbf{r}_{t+1} \mid \mathbf{z}_{t+1}) \\ +D_{KL}(q_\psi(\mathbf{z}_{t+1} \mid \mathbf{x}_{t+1}, \mathbf{z}_t, \mathbf{a}_t) \, p_\psi(\mathbf{z}_{t+1} \mid \mathbf{z}_t, \mathbf{a}_t)) \end{array} \right]. \tag{4}$$

The resulting inferred latent state, which captures information about previous observations and expected reward in future time steps, serves as the input for the critic in SAC.

In practice, the probability distributions in Equations 2 and 3 are set to follow Gaussian distributions with diagonal covariance matrices, where the means and variances are the output of neural networks, optimized using the reparametrization trick (Kingma & Welling, 2014). The neural network that models $q_\psi(\mathbf{z}_1^1 \mid \mathbf{x}_1)$ treats a lower-dimensional feature vector, extracted from the high-dimensional observation using a convolutional neural network. For further details, we refer the interested reader to Lee et al. (2020).

We can solve this problem taking a Lagrangian relaxation of the constrained problem and optimizing the unconstrained dual problem through dual gradient descent (Geibel & Wysotzki, 2005; Stooke et al., 2020). SAC-Lagrangian uses this approach, to bring SAC to the safety-constrained CMDP setting, where it may constrain the step-wise cost (Ha et al., 2020) or cost return (Yang et al., 2021). In the next section, we present our approach to adapt SLAC for the CPOMDP setting.

## 5 SAFE SLAC

The Safe SLAC model is an adaption of SLAC for the safety-constrained setting under partial observability. There are three steps to this adaption. First, we adapt the latent variable model to include information related to safety in the latent state. Second, we train a safety critic to predict the expected cost return resulting from a given action. Third, we constrain the policy objective and use its Lagrangian relaxation to train the actor.

**Latent variable model with cost prediction.** We include a notion of safety in the latent state by changing the generative part of the model to predict the cost at the next time step, in addition to predicting the observation and reward. This introduces the conditional probability distribution $c_t \sim p_{\psi^c}(c_t \mid \mathbf{z}_t, \mathbf{a}_t, \mathbf{z}_{t+1})$ to the generative component of the model, analogous to the component that predicts reward. Using the simplified notation of the parameters ($\psi$), the resulting model objective is:

$$J_M(\psi) = \mathop{\mathbb{E}}_{\mathbf{z}_{1:\tau+1} \sim q_\psi} \left[ \sum_{t=0}^{\tau} \begin{array}{l} -\log p_\psi(\mathbf{x}_{t+1}|\mathbf{z}_{t+1}) - \log p_\psi(\mathbf{r}_{t+1}|\mathbf{z}_{t+1}) - \log p_\psi(\mathbf{c}_{t+1}|\mathbf{z}_{t+1}) \\ +D_{KL}(q_\psi(\mathbf{z}_{t+1} \mid \mathbf{x}_{t+1}, \mathbf{z}_t, \mathbf{a}_t) \parallel p_\psi(\mathbf{z}_{t+1} \mid \mathbf{z}_t, \mathbf{a}_t)) \end{array} \right]. \tag{5}$$

While the other probability distributions in the model are Gaussian variables with a diagonal covariance matrix, we follow As et al. (2022) and use a Bernoulli distribution for the conditional cost distribution, since the cost is binary in the environments we consider.

**Safety critic.** As an integral part of Safe SLAC, we train a safety critic. First, the reward critic $Q^r$ with parameters $\theta$ remains unchanged from SLAC and estimates the expected discounted reward return for a given latent state and action pair under the current policy, as well as the expected entropy of the policy after taking the given action in the environment. To this end, $Q^r$ optimizes the soft Bellman residual in Equation 6.

$$J_{Q^r}(\theta) = \mathop{\mathbb{E}}_{\mathbf{z}_{1:\tau+1} \sim q_\psi} \left[ \frac{1}{2} (Q_\theta^r(\mathbf{z}_\tau, \mathbf{a}_\tau) - (r_\tau + \gamma V_{\bar{\theta}}(\mathbf{z}_{\tau+1})))^2 \right]. \tag{6}$$

$$V_\theta(\mathbf{z}_{\tau+1}) = \mathop{\mathbb{E}}_{\mathbf{a}_{\tau+1} \sim \pi_\phi} \left[ Q_\theta^r(\mathbf{z}_{\tau+1}, \mathbf{a}_{\tau+1}) - \alpha \log \pi_\phi(\mathbf{a}_{\tau+1} \mid \mathbf{x}_{1:\tau+1}, \mathbf{a}_{1:\tau}) \right]. \tag{7}$$

In addition to the reward critic, we train the safety critic $Q^c$ with parameters $\zeta$ that predicts the expected discounted cost return for a given latent state and action. The loss function for the safety critic, given in Equation 8, is similar to Equation 6 but does not include an entropy term.

$$J_{Q^c}(\zeta) = \mathop{\mathbb{E}}_{\mathbf{z}_{1:\tau+1} \sim q_\psi} \left[ \frac{1}{2} (Q_\zeta^c(\mathbf{z}_\tau, \mathbf{a}_\tau) - (c_\tau + \gamma \mathop{\mathbb{E}}_{\mathbf{a}_{\tau+1} \sim \pi_\phi} Q_{\bar\zeta}^c(\mathbf{z}_{\tau+1}, \mathbf{a}_{\tau+1})))^2 \right]. \tag{8}$$

**Safe SLAC policy objective.** SLAC follows the maximum entropy framework objective (Haarnoja et al., 2018b), using the latent state as input for the critics. We adjust this to be constrained to safe behavior. The policy objective in safety-indifferent SLAC maximizes both the entropy of the policy, which ensures sufficient exploration, and the soft action-value estimate as predicted by the reward critic $Q_\theta^r$:

$$J_\pi(\phi) = \mathop{\mathbb{E}}_{\mathbf{z}_{t:\tau+1} \sim q_\psi} \left[ \mathop{\mathbb{E}}_{\mathbf{a}_{\tau+1} \sim \pi_\phi} \left[ \alpha \log \pi_\phi(\mathbf{a}_{\tau+1} \mid \mathbf{x}_{1:\tau+1}, \mathbf{a}_{1:\tau}) - Q_\theta^r(\mathbf{z}_{\tau+1}, \mathbf{a}_{\tau+1}) \right] \right]. \tag{9}$$

Analogously to the safety critic term in SAC-Lagrangian (Ha et al., 2020), we obtain the objective for the safety-constrained policy by including a term to minimize the expected cost return as estimated by the safety critic $Q_\zeta^c$, weighted by the Lagrange multiplier $\lambda$:

$$J_\pi(\phi) = \mathop{\mathbb{E}}_{\substack{\mathbf{z}_{t:\tau+1} \sim q_\psi \\ \mathbf{a}_{\tau+1} \sim \pi_\phi}} \left[ \alpha \log \pi_\phi(\mathbf{a}_{\tau+1} \mid \mathbf{x}_{1:\tau+1}, \mathbf{a}_{1:\tau}) - Q_\theta^r(\mathbf{z}_{\tau+1}, \mathbf{a}_{\tau+1}) + \lambda Q_\zeta^c(\mathbf{z}_{\tau+1}, \mathbf{a}_{\tau+1}) \right]. \tag{10}$$

As we discussed in Section 3, it is not practical to condition the policy on the full history. To circumvent this problem, Lee et al. (2020) use a truncated history of observations and actions as input for the policy; however, we find empirically that using the latent state as input for the actor yields better results in our setting. As such, in our training procedure the policy takes as input the current latent state $\mathbf{z}$, which is updated after each environment step. When using the latent state as input for the policy, occurrences of $\pi_\phi$ in Equations 6, 9 and 10 and Algorithm 1 can instead be read as modeling the distribution $\pi_\phi(\mathbf{a}_{\tau+1} \mid \mathbf{z}_{\tau+1})$ and $\pi_\phi(\mathbf{a}_t \mid \mathbf{z}_t)$, respectively. In the gradient steps, we infer the latent state from a truncated history of observations and actions that is sampled from the replay buffer, rather than from the full history.

Note that Safe SLAC carries the same limitation as SLAC. Overall, even in the unconstrained case, conditioning the policy on the latent variable is not guaranteed to result in the optimal policy of the underlying POMDP (see Lee et al., 2020). To find an optimal policy, we must condition the policy in the history or a belief state that provides sufficient statistics of the history, which may lead the agent to take actions to gather information (Cassandra et al., 1994; Thrun, 1999).

The entropy and expected cost-return are constrained, therefore, $\alpha$ and $\lambda$ are adjusted dynamically to incentivize the policy to adhere to the constraints. $\alpha$ is tuned using the procedure and hyperparameters as in (Haarnoja et al., 2018b). Next, we describe how to adjust $\lambda$ during training.

### 5.1 Safety Lagrange multiplier learning

We use on-policy data to adjust the Lagrange safety multiplier. Previous work in the fully observable setting has adjusted the Lagrange multiplier by estimating the cost return as the mean prediction of the safety critic over states sampled from the replay buffer(Yang et al., 2021). It has been shown for Q-learning (Thrun & Schwartz, 1993) and policy gradient methods (Fujimoto et al., 2018), that using function approximators such as neural networks to estimate state and action value functions can result in biased value estimates. While bias in value estimation can result in suboptimal policy learning, the induced bias is doubly important if the safety critic not only serves to provide a gradient to the agent, but its magnitude is also used to determine the Lagrange multiplier. We have found in our experiments that for the given environments, this bias was sufficient to prevent the learning of safe policies and we were unable to mitigate this using the techniques proposed by Fujimoto et al. (2018). We attribute this to the high degree of partial observability in the environments, the off-policy nature of the learning algorithm, and the use of 0-step temporal difference error for value updating. Consequently, we adjust the Lagrange multiplier $\lambda$ based on the real undiscounted cost return incurred in each training episode. This is expressed in the following loss, where $\lambda$ is changed to minimize the incurred cost return less the allowed cost budget:

$$J_s(\lambda) = \mathop{\mathbb{E}}_{\rho \sim \rho_\pi} \left[ \lambda \sum_{t=1}^{T} c_t - d \right], \tag{11}$$

---

**Algorithm 1** Safe SLAC

---

**Input**: Empty replay buffer $\mathcal{D}$, environment $\mathcal{E}$, and initialized parameters $\psi$, $\theta_1$, $\theta_2$, $\zeta$, and $\phi$
**Hyperparameter**: Number of warmup interactions $W_p$ and warmup training steps $W_t$
**Output**: Optimized parameters

1: Sample $W_p$ transitions from $\mathcal{E}$ using a stochastic policy and store these in $\mathcal{D}$.
2: **for** i=1 to $W_t$ **do**
3:     Sample experience from $\mathcal{D}$
4:     Update $\psi$ according to Equation 5
5: **end for**
6: **while** not converged **do**
7:     **for** each environment step at time $t$ **do**
8:         $\mathbf{a}_t \sim \pi_\phi(\mathbf{a}_t \,|\, \mathbf{x}_{1:t}, \mathbf{a}_{1:t-1})$
9:         Obtain $(\mathbf{x}_{t+1}, r_{t+1}, c_{t+1})$ by executing $\mathbf{a}_t$ in $\mathcal{E}$
10:        $\mathcal{D} \leftarrow \mathcal{D} \cup (\mathbf{x}_{t+1}, \mathbf{a}_t, r_{t+1}, c_{t+1})$
11:        Update $\lambda$ by minimizing Equation 11
12:     **end for**
13:     **for** each gradient step **do**
14:         $(\mathbf{x}_{1:\tau+1}, \mathbf{a}_{1:\tau}, r_{1:\tau+1}, c_{1:\tau+1}) \sim \mathcal{D}$
15:        Update $\theta_1, \theta_2$ according to Equation 6
16:        Update $\psi$ according to Equation 5
17:        Update $\phi$ according to Equation 10
18:        Update $\zeta$ according to Equation 8
19:        $\bar{\theta}_1 \leftarrow \nu\theta_1 + (1 - \nu)\bar{\theta}_1$
20:        $\bar{\theta}_2 \leftarrow \nu\theta_2 + (1 - \nu)\bar{\theta}_2$
21:        $\bar{\zeta} \leftarrow \nu\zeta + (1 - \nu)\bar{\zeta}$
22:     **end for**
23: **end while**

---

where $\rho$ is a trajectory and $d$ is the budget. Intuitively, the weight of the safety critic in the policy loss is increased when the policy is unsafe and decreased when the policy is safe. In practice, we found this procedure to be stable and effective for a sufficiently low learning rate. We stress that the evaluation episodes are not used for policy learning, including adjustment of the Lagrange multiplier.

## 5.2 TRAINING PROCEDURE

Safe SLAC is described in Algorithm 1. The training procedure starts initializing the replay buffer by taking actions under a stochastic policy for 60k environment steps, with actions sampled from a truncated diagonal Gaussian distribution. It then trains an initial latent variable model for 30k steps on sequences uniformly sampled from the replay buffer, minimizing the objective in Equation 5. After this warm-up procedure, we alternate between collecting experiences and training the latent variable model, actor, and critics.

To improve stability during training, we maintain target networks for the critics, which are updated using exponential averaging (Lines 19 to 21). The neural networks that parameterize the model, actor, and critics have the same architecture as in SLAC, with hyperparameters as detailed in Appendix B. Except for the safety Lagrange multiplier, we optimize all parameters using the Adam optimizer (Kingma & Ba, 2014).

Whereas Lee et al. (2020) indicate that the optimal additive noise added to the observation prediction during latent variable model training is of low variance for most environments, we find better results with a large variance (0.4). This is consistent with their hypothesis that a strong noise yields better results for environments with large differences between subsequent observations, as is the case in our environments.

We follow both SLAC and LAMBDA in using two action repeats on these environments, meaning that we transform the base environment so that each action executed in it is performed twice before the agent receives a new observation. We aggregated both the reward and cost incurred in the transitions induced by these repeated actions to scalar values by summing them.

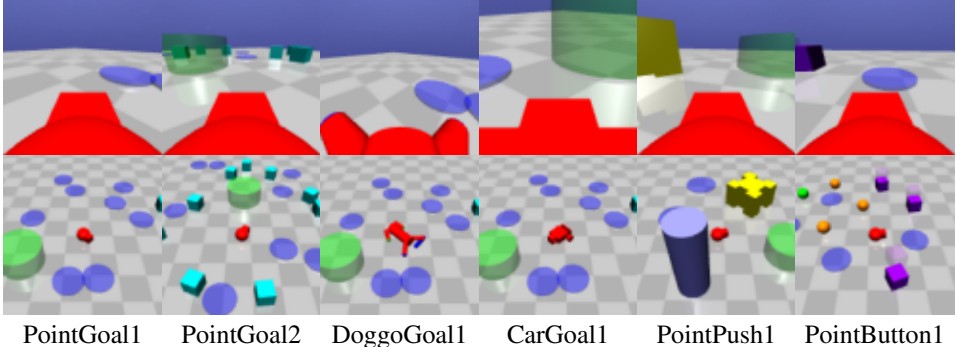

PointGoal1    PointGoal2    DoggoGoal1    CarGoal1    PointPush1    PointButton1

Figure 1: Six SG6 environments used in our experiments. Top row shows the first-person camera pixel image that our agent observes. Bottom row gives a top-down overview of the environment. In all environments, the agent (red) must reach the goal (green) while avoiding the hazards (blue and purple). In PointPush1, the agent must push the yellow box into the green goal to receive a reward.

## 6 EMPIRICAL EVALUATION

We evaluate our approach on a set of six SafetyGym benchmark environments introduced by Ray et al. (2019) as SG6, shown in Figure 1. For all environments, we use the standard cost return budget of 25 per episode. As training progresses, we frequently evaluate the performance of the policy in 10 evaluation episodes. The mean reward and cost return during these evaluation episodes are shown in Figure 2 for three different random seeds per environment. We calculate the normalized reward and cost return using the performance in the last 5 evaluations, as proposed by Ray et al. (2019). We compare our results with the final results from basic algorithms, including the unconstrained proximal policy optimization (PPO; Schulman et al., 2017), using sensor observations, and with LAMBDA trained on pixel observations, as reported by As et al. (2022); see Section 2 for a description of these algorithms. Figures 2 and 3 compile the results obtained in the SG6 set. Appendix D presents a qualitative analysis of the results.

Figure 2 shows that Safe SLAC converges on a safe policy for nearly all environments and attains a competitive reward return. Safe SLAC reaches a higher reward return than LAMBDA compared to other baselines such as PointGoal1 and CarGoal1 environments. In PointGoal2, we also reach a higher reward return than LAMBDA, but only after 2 million environment steps. The results shown for LAMBDA in PointGoal2 used training for only 1 million steps; we reach the same performance as LAMBDA using approximately 40% more samples and continue to improve after this.

**Safety and reward tradeoff.** In the DoggoGoal1 environment, we observe that while the reward return of Safe SLAC is significantly better than for the other algorithms, the mean cost return is slightly larger than the allowed limit and is only better than CPO. This indicates that the tradeoff between reward and safety is balanced too far towards reward. Adjusting the learning rate for the safety Lagrange multiplier from the default $2 * 10^{-6}$ to $6 * 10^{-6}$, we obtain the results indicated in Figure 3 as Safe SLAC+, which increases the reward return in PointGoal2. From this effect, we conclude that our policy makes a meaningful tradeoff between optimizing reward and cost return, and that the current training procedure for the Lagrange multiplier is sensitive to the learning rate.

**Challenges in strong partial-observability.** Like LAMBDA, our approach fails to make substantial improvement in reward return in PointPush1, although the policy is safe. We attribute this to the strong partial observability due to the pixel observations: while the box is pushed, the agent cannot observe its surroundings and so must rely entirely on its belief or latent state. This is supported by the fact that the baselines that use sensor observations are able to attain a much higher reward return.

**Normalized metrics.** To summarize the performance of our algorithm, we normalize the mean reward return and cost return for each environment at the end of training by dividing them by the performance of unconstrained PPO, as proposed by Ray et al. (2019). The resulting normalized metrics, shown in Figure 3, indicate that Safe SLAC improves over the baselines and is competitive

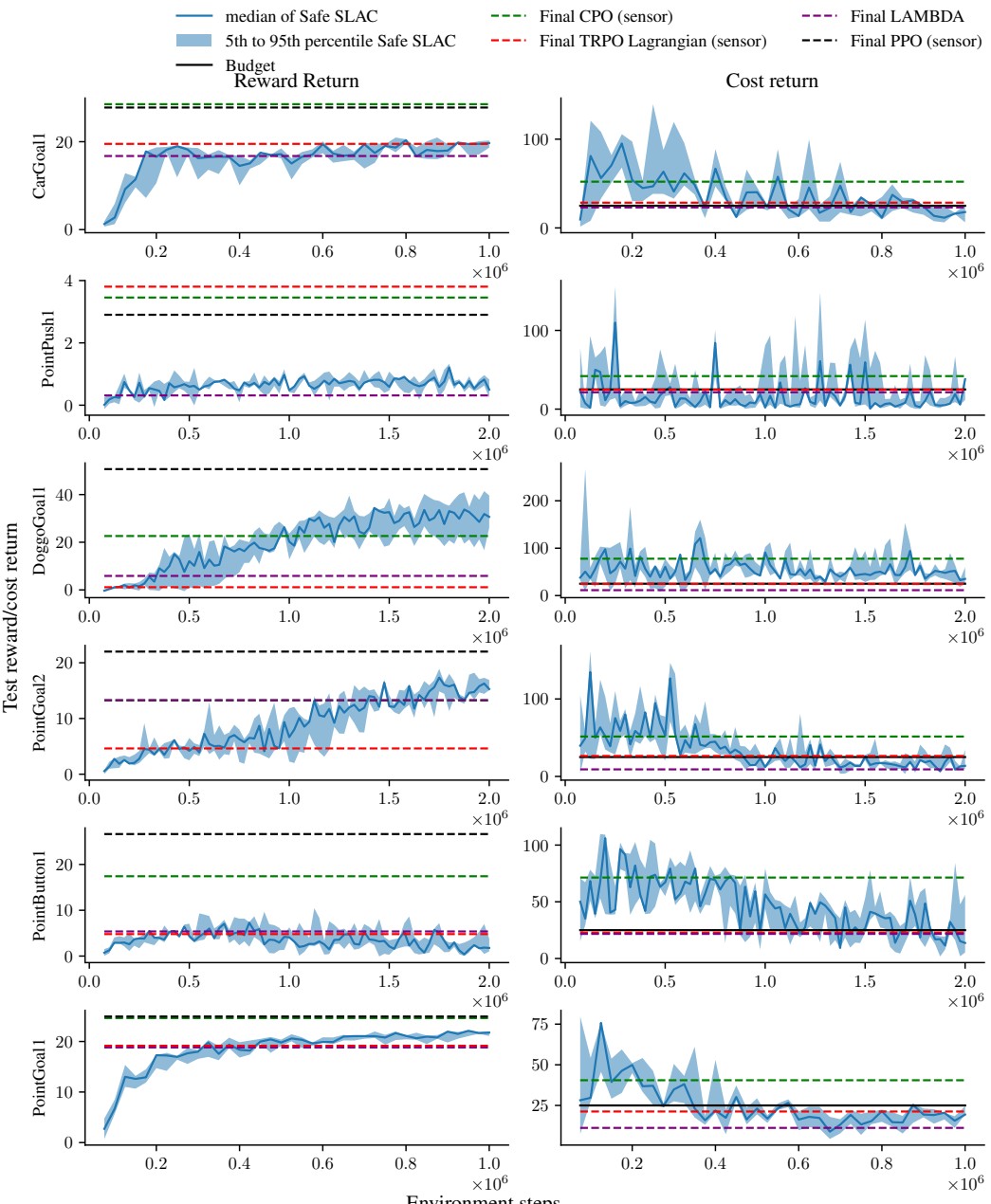

Figure 2: Evaluation reward and cost return. Note that for PointGoal2, the final results for LAMBDA are calculated after 1 million environment steps, whereas we use 2 million.

with LAMBDA. We exclude DoggoGoal1 from this figure, as Safe SLAC is not safe for our standard learning rate as discussed above. The normalized reward return is so large for Safe SLAC that it obscures the performance in other environments. The same figure with DoggoGoal1 included can be seen in Figure 5 (Appendix C).

**History as actor input.** As discussed in Section 5, we deviate from the suggestion by Lee et al. (2020) to use the truncated history as input for the actor and instead use the latent state inferred from the full history to select actions. In Figure 4, we compare the performance of these two policy inputs on the PointGoal2 environment. While both variants converge to a safe policy, the reward performance of the latent state policy is higher than that of the history policy.

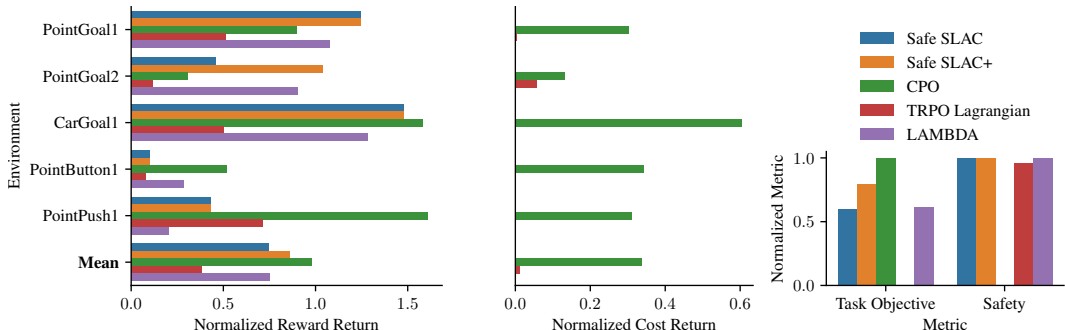

Figure 3: Normalized reward return and normalized cost return performance of Safe SLAC compared to LAMBDA and the baselines. "Safe SLAC" indicates our approach evaluated using a single set of hyperparameters for all environments and after 1 million environments steps on PointGoal2, to match LAMBDA. "Safe SLAC+" shows results using an adjusted learning rate on the DoggoGoal1 environment and 2 million steps on PointGoal2.

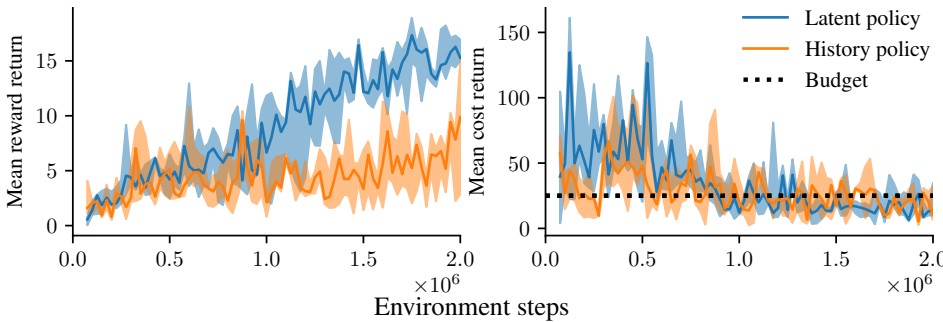

Figure 4: Comparison of latent state and truncated history as actor input on the PointGoal2 environment. While both are safe at the end of training, the latent state policy shows higher reward return.

**Safe SLAC may be less pessimistic than LAMBDA.** For PointGoal2, LAMBDA was trained only with 1M environment steps. For this computational budget, Safe SLAC is able to find a safe policy, however, we notice it has not converged yet, and the reward return is slightly lower than the final reward return from LAMBDA. Nevertheless, with more time Safe SLAC clearly outperforms LAMBDA, whose reward return in this environment plateaus before reaching 1 million environment steps (Figure 5 from As et al., 2022). In summary, Safe SLAC can keep improving while LAMBDA stagnates as we use more of the allowed cost budget.

**Computational requirements.** Our approach is less complex than LAMBDA, as it does not require costly sampling of Bayesian world models, or generating synthetic experience using the latent variable model. Thus, we conjecture that Safe SLAC also has lower computational requirements. As an example, training our approach for 1 million base environment steps on PointGoal1 takes a total of 10 hours and 14 minutes. For contrast, LAMBDA takes 13 hours and 20 minutes (measured on the same hardware). Nevertheless, this improvement might also be due to implementations details.

## 7 CONCLUSION

This paper proposes Safe SLAC, a safety-constrained RL approach for partially observable settings, which uses a stochastic latent variable model combined with a safety critic. Empirically, it shows competitive performance with state-of-the-art methods while having a less complex training procedure. Safe SLAC often derives policies with higher reward returns, while still satisfying the safety constraint. Future work includes using a distributed critic to increase the control over the safety violations (Yang et al., 2022) and improving the optimization of the Lagrange multiplier (Stooke et al., 2020), which is currently sensitive to its learning rate as seen in the DoggoGoal1 experiments.

ACKNOWLEDGMENTS

This work was funded by the ERC Starting Grant 101077178 (DEUCE) and the NWO grant NWA.1160.18.238 (PrimaVera). We thank Toshiki Watanabe for their reimplementation of SLAC in PyTorch, which served as the basis for our code.

REPRODUCIBILITY STATEMENT

The code for our implementation of Safe SLAC is available on GitHub at `https://github.com/lava-lab/safe-slac`. The repository contains implementations for the Safe SLAC actor, critics, model and training procedure. This includes the necessary code to interface with the publicly available SafetyGym environments using pixel observations, as well as instructions to install the required dependencies and reproduce our experiments. In addition, the default hyperparameters are specified in a configuration file.

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

## A  LATENT VARIABLE MODEL

Following the observations from Lee et al. (2020, Appendix B), we implemented a model composed of two latent variables $\mathbf{z}^1$ and $\mathbf{z}^2$. Equations 12 and 13 provide the detailed description of this model.

$$\mathbf{z}_1^1 \sim p(\mathbf{z}_1^1). \tag{12a}$$

$$\mathbf{z}_1^2 \sim p_{\psi^{z02}}(\mathbf{z}_1^2 \mid \mathbf{z}_1^1). \tag{12b}$$

$$\mathbf{z}_{t+1}^1 \sim p_{\psi^{z1}}(\mathbf{z}_{t+1}^1 \mid \mathbf{z}_t^2, \mathbf{a}_t). \tag{12c}$$

$$\mathbf{z}_{t+1}^2 \sim p_{\psi^{z2}}(\mathbf{z}_{t+1}^2 \mid \mathbf{z}_{t+1}^1, \mathbf{z}_t^2, \mathbf{a}_t). \tag{12d}$$

$$\mathbf{x}_t \sim p_{\psi^x}(\mathbf{x}_t \mid \mathbf{z}_t^1, \mathbf{z}_t^2). \tag{12e}$$

$$\mathbf{r}_t \sim p_{\psi^r}(\mathbf{r}_t \mid \mathbf{z}_t^1, \mathbf{z}_t^2, \mathbf{a}_t, \mathbf{z}_{t+1}^1, \mathbf{z}_{t+1}^2). \tag{12f}$$

$$\mathbf{z}_1^1 \sim q_{\psi^{z01}}(\mathbf{z}_1^1 \mid \mathbf{x}_1). \tag{13a}$$

$$\mathbf{z}_1^2 \sim p_{\psi^{z02}}(\mathbf{z}_1^2 \mid \mathbf{z}_1^1). \tag{13b}$$

$$\mathbf{z}_{t+1}^1 \sim q_{\psi^{z1}}(\mathbf{z}_{t+1}^1 \mid \mathbf{x}_{t+1}, \mathbf{z}_t^2, \mathbf{a}_t). \tag{13c}$$

$$\mathbf{z}_{t+1}^2 \sim p_{\psi^{z2}}(\mathbf{z}_{t+1}^2 \mid \mathbf{z}_{t+1}^1, \mathbf{z}_t^2, \mathbf{a}_t). \tag{13d}$$

Notice that in this factorization the generative and variational models share some parts, namely Equations 13b and 13d are the same as Equations 12b and 13d, respectively. As Lee et al. (2020) shows, this allow us to compute the KL-divergence $D_{KL}(q_\psi(\mathbf{z}_{t+1} \mid \mathbf{x}_{t+1}, \mathbf{z}_t, \mathbf{a}_t) \parallel p_\psi(\mathbf{z}_{t+1} \mid \mathbf{z}_t, \mathbf{a}_t))$ based only on the $\mathbf{z}^1$, which simplifies the training process. We refer the reader to Lee et al. (2020) for further details of the implementation.

## B  HYPERPARAMETERS

Table 1: Hyperparameters used for safe SLAC

| Parameter | Value |
|---|---:|
| Action repeat | 2 |
| Image size | $64 \times 64 \times 3$ |
| Image reconstruction noise | 0.4 |
| Length of sequences sampled from replay buffer | 10 |
| Discount factor | 0.99 |
| Cost discount factor | 0.995 |
| $z^1$ size | 32 |
| $z^2$ size | 200 |
| Replay buffer size | $2 * 10^5$ |
| Latent model update batch size | 32 |
| Actor-critic update batch size | 64 |
| Latent model learning rate | $1 * 10^{-4}$ |
| Actor-critic learning rate | $2 * 10^{-4}$ |
| Safety Lagrange multiplier learning rate | $2e - 4$ |
| Initial value for $\alpha$ | $4 * 10^{-3}$ |
| Initial value for $\lambda$ | $2 * 10^{-2}$ |
| Warmup environment steps | $60 * 10^3$ |
| Warmup latent model training steps | $30 * 10^3$ |
| Gradient clipping max norm | 40 |
| Target network update exponential factor | $5 * 10^{-3}$ |

## C NORMALIZED RESULTS INCLUDING DOGGOGOAL1

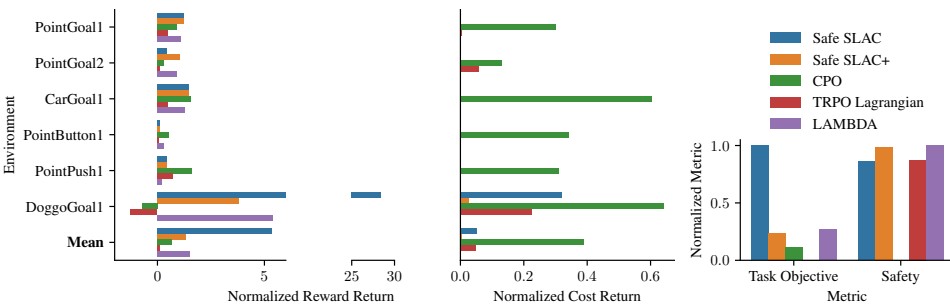

Figure 5: Normalized reward return and normalized cost return performance of Safe SLAC compared to LAMBDA and the baselines. "Safe SLAC" indicates our approach evaluated using a single set of hyperparameters for all environments and after 1 million environments steps on PointGoal2, to match LAMBDA. "Safe SLAC+" shows results using an adjusted learning rate on the DoggoGoal1 environment and 2 million steps on PointGoal2.

## D QUALITATIVE ANALYSIS OF THE RESULTS

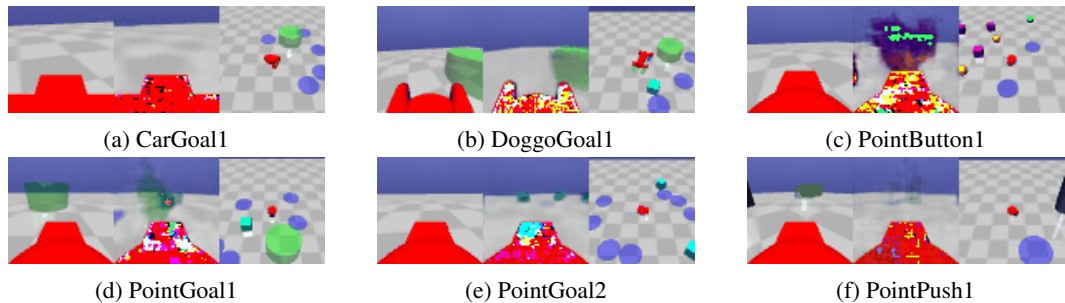

Figure 6: Sample from the roll-outs of the final policies in each environment.

Using the final policy computed by Safe SLAC, we collected a few trajectories in each environment to make a qualitative analysis. Figure 6 shows one time-step for each environment. For each environment, we have three videos showing the observation, the latent variable model reconstruction, and an overview of the environment.[1]

We observe that Safe-SLAC can effectively solve the PointGoal1, PointGoal2, and CarGoal1 tasks. In these environments, the agent has an exploration strategy of turning 180 degrees in place when the target is not visible. In PointGoal2, we also see the agent carefully crossing spaces between hazards. As expected, the agent mostly moves forward in these environments, while the agents using sensor observations also move backward. [2]

In the PointButton1 environment, we observe that the agent can reach the target a few times but might fail when the goal is not in the field of vision of the agent.

In the DoggoGoal1 environment, the agent can also reach the goal. However, the spaces between the hazards can be small compared to the robot's size. We suspect this makes this task considerably more complex than the remaining tasks in terms of safety.

Finally, in the PointPush1 environment, the agent goes to the box. However, it does not move the box to the goal position. This is due to the strong partial observability of this environment since the agent cannot observe the goal when the box is between the agent and the goal position.

---

[1]Videos are available at https://github.com/lava-lab/safe-slac#sample-rollouts.
[2]https://openai.com/blog/safety-gym/

