# OpenReview forum: "Safe Reinforcement Learning From Pixels Using a Stochastic Latent Representation"
_ICLR.cc/2023/Conference — ICLR 2023 poster_

### Official Review · Reviewer_4rUn · 2022-10-19

**Confidence:** 4
**Correctness:** 3
**Technical Novelty And Significance:** 2
**Empirical Novelty And Significance:** 2
**Recommendation:** 6

**Clarity, Quality, Novelty And Reproducibility:**

* The authors point out that the pessimistic ensembling techniques used in LAMBDA [2] might be overly pessimistic. While I agree that this is a risk, I would like to point out that LAMBDA is a model-based algorithm where policy and critic are trained on model rollouts. These will have some error, and the pessimism might be necessary in that setup. SLAC, on the other hand uses model-free RL (with the exception of the latent representation) and pessimistic/optimistic approximations might simply not be necessary here. The performance gap between safe SLAC and LAMBDA appears quite small and I don't see a strong evidence to support the claim that "Safe SLAC is less pessimistic than LAMBDA".
* Regarding the performance gap between safe SLAC and LAMBDA, I'm not sure it is a good decision to use Figure 3. It is somewhat difficult to read due to the outlier caused by DoggoGoal1. Figure 5 is easier to read in comparison, though at the cost of removing DoggoGoal1 from the benchmark. Perhaps the authors could give the reader some intuition about the policy's behavior. You are saying that the policy achieves a very high reward by acting unsafely. What does this mean here? Does it completely ignore constraints in a way that would lead to hazardous behavior in a real-life setup? The point is, either the policy's behavior is acceptable or it is not. If it is not, then it shouldn't appear in the comparison, as it makes it look like safe SLAC has a huge performance edge over LAMBDA. If it is, then you should point out why. If it is difficult to judge whether the policy behaves acceptably, I would advise using Figure 5 instead of Figure 3.
* The authors suggest that using the latent state instead of a history of interactions as input to the policy works better. I can completely see why this would be the case, though I think it should be pointed out that this strategy will not work on any given POMDP, since some POMDPs require the agent to learn to actively acquire new information (e.g. the heaven and hell task from [3]), which is only possible by working with interaction histories (or sufficient statistics thereof) or belief states, which the latent state does not necessarily substitute. I don't see this as a huge issue, since POMDPs come in a large variety, and using the latent state will be a viable approach for many of those, but this shortcoming should be addressed in the text.

---

[2] Constrained Policy Optimization via Bayesian World Models, https://arxiv.org/pdf/2201.09802.pdf

[3] Monte Carlo POMDPs, https://papers.nips.cc/paper/1999/file/299570476c6f0309545110c592b6a63b-Paper.pdf

**Strength And Weaknesses:**

**strengths**

* The paper proposes a method for safe RL in POMDPs which is competitive with the state-of-the-art.
* The method mostly consists of previously established work that is combined in a new way. I see this as a strength.

**weaknesses**

* The performance gap to previous work is not clear. As such, the paper is not an improvement over the current state of this setup. This is my main reason for the score I've given to the paper.

**Summary Of The Paper:**

This paper extends SLAC [1] to make it applicable to constrained POMDPs. The proposed approach combines model-free RL with a latent representation that is learned by a generative model. Constraints are satisfied by learning a constraint critic and optimizing a Lagrangian, which boils down to extending the regular SAC objective by another critic component that is weighted by a Lagrange multiplier. The method performs on par with the state-of-the-art on the safety-gym benchmark

---

[1] Stochastic Latent Actor-Critic: Deep Reinforcement Learning with a Latent Variable Model, https://arxiv.org/pdf/1907.00953.pdf

**Summary Of The Review:**

This paper is an iteration over SLAC, which modifies that work with a constraint critic, allowing application to constrained POMDPs. The method consists mostly of existing approaches combined in a new form, which has the benefit that each component has a history of related work supporting it. That being said, the method does not have a clear edge over the closest related work (LAMBDA). Thus, I see this paper as a new approach for attaining state-of-the-art performance in a benchmark for safe-RL. I believe this is still a contribution worthy of acceptance, since it will widen researchers' knowledge of which methods work and which do not.

---

> ### Author Response · Authors · 2022-11-11
> **Author Response**
>
> Dear reviewer 4rUn, thank you for the thoughtful comments and the constructive criticism.
>
> > The performance gap between safe SLAC and LAMBDA appears quite small and I don't see a strong evidence to support the claim that "Safe SLAC is less pessimistic than LAMBDA"
>
>
> Regarding the pessimism of LAMBDA, we agree the difference in performance is most significant in the PointGoal2 environment. However, analyzing the Figure 3 (in the revised document),we observe that Safe SLAC+ also outperforms LAMBDA in multiple environments, namely CarGoal1, PointPush1, and PointGoal1 although with a smaller margin. Nevertheless, we decided to softened our statement to "Safe SLAC may be less pessimistic than LAMBDA."
>
>
> > The performance gap to previous work is not clear.
>
> Indeed, we observe that Safe SLAC can outperform LAMBDA in some environments given the hyper-parameters used, while it still requires more hyper-parameter tuning in others. Nevertheless, as the reviewer mentioned, Safe SLAC is "a new approach for attaining state-of-the-art performance in a benchmark for safe-RL." Where LAMBDA introduces complexity in both implementation and computational requirements by sampling from a Bayesian world model to generate millions of synthetic environment interactions per episode, our approach learns directly from past experiences. We consider this a significant advance since Safe SLAC uses an approach considerably simpler than prior work, therefore easier to implement and benchmark.
>
> > I would advise using Figure 5 instead of Figure 3
>
> Thanks for the thoughtful comments regarding the performance of Safe SLAC in the DoggoGoal1 environment. To reduce the effects of this outlier, we have swapped figures 3 and 5 as suggested and updated the discussion accordingly.
>
> > Perhaps the authors could give the reader some intuition about the policy's behavior.
>
> Taking that into account, we performed a more careful qualitative analysis of the policies computed by Safe SLAC in each environment (Appendix D). The videos are available at the anonymous github repository https://github.com/safe-slac/safe-slac#sample-rollouts
>
> > this shortcoming should be addressed in the text
>
> Indeed, Safe SLAC carries the same limitations from the SLAC algorithm when dealing with a POMDP. We added this discussion to the Safe SLAC section in the paragraph "Sufficient statistics." In summary, SLAC is not able to take actions that reduce the state uncertainty, therefore, it may fail to solve some POMDPs.
>
> Please let us know if this addressed your concerns or if you have further questions, we would be happy to answer them.

---

### Official Review · Reviewer_asV6 · 2022-10-25

**Confidence:** 3
**Correctness:** 3
**Technical Novelty And Significance:** 3
**Empirical Novelty And Significance:** 3
**Recommendation:** 8

**Clarity, Quality, Novelty And Reproducibility:**

The paper is well-written. Furthermore, the presented modifications to the SLAC framework onto the safety-constrained setting are straightforward, making the method practical. Equations (2) and (3) along with the preceding paragraph could be more explicit. For example, what do the superscripts of the latent variables z_i^j mean?

**Strength And Weaknesses:**

The trained policy is empirically evaluated on a set of challenging, fairly-recent safety-gym environments (Ray et al. 2019). The method seems to learn safe, competitive policies in most of the environments (with DoggoGoal1and PointPush as exceptions). However, it is difficult to understand whether these policies are any good. A potential quick fix here is to plot a line similar to the budget line on the cost return plots but for the policy. This line would represent the reward return for the worst policy that still solves the problem. Later in the paper, there is a discussion on computational requirements where the running time of the method is compared to that of LAMBDA (As et al. 2022). However, it is not clear to the reader whether or not this difference was due to implementation details or computational complexity.

**Summary Of The Paper:**

This paper proposes an approach for safe reinforcement learning from pixel observations. This is framed as a POMDP problem where the agent receives separate cost signals from the environment in addition to rewards. At a high-level, this work adapts the Stochastic Latent Actor-Critic framework (which learns a latent representation along with a dynamics model in the latent space) to handle safety constraints. Particularly, soft constraints since the agent can violate them at every time step for a cost. Thus the resulting learned policy becomes "safe" by minimizing some notion of expected cumulative cost in addition to maximizing the expected return. In this work, safety is considered through the addition of a separate safety critic in the policy objective that predicts the expected cost return given a latent state and an action.

**Summary Of The Review:**

This paper presents a straightforward set of modifications to the SLAC framework onto the safety-constrained setting. This would make implementation and benchmarking fairly simple since the environments used are also standard in this setting.

---

> ### Author Response · Authors · 2022-11-11
> **Author Response**
>
> Dear reviewer asV6, thank you for the thorough review and constructive criticism. We are glad to read that the paper is well-written and the approach is practical.
>
> > it is difficult to understand whether these policies are any good
>
>
> Thanks for bringing this to our attention. To clarify whether the policies are solving the tasks or not, we included the performance of an unconstrained agent (namely vanilla PPO) to the plots in Figure 2. This is compatible with the approach of comparing the results with PPO in the normalized results proposed in the safety gym benchmark (Ray et al. 2019).
> Furthermore, we observe in the videos attached to the code (https://github.com/safe-slac/safe-slac), that in most environments the Safe SLAC computes policies that perform the task well. We added a more extensive discussion in Appendix D.
>
> > Equations (2) and (3) along with the preceding paragraph could be more explicit.
>
> We made several improvements on the presentation of equations 2 and 3. In particular, we now have a simplified version of the latent variable model in the main text and a in-depth version in the supplemental material.
>
> > superscripts of the latent variables
>
>
> The superscripts of the latent variable are related to the factorization of the latent variable model proposed by Lee et al. (2020). To improve the presentation, we simplified the model of the main document and added the factorized model in the supplemental material, including an appropriate reference in the main document.
>
> >  implementation details or computational complexity
>
> Thanks for raising this issue. This is indeed a point that must be treated carefully.
> We agree that it is difficult to conclude whether the difference in runtime is due to implementation details or fundamental differences in complexity. We have expanded this part of the discussion in the paper to more accurately reflect this issue.
>
> We hope to have resolved your concerns. Please let us know if you have any further questions or comments, we will be happy to address them within the discussion period.

---

> > ### Comment · Reviewer_asV6 · 2022-12-01
> > **Response to the authors' response**
> >
> > Thank you for addressing my concerns. In the latest revision of the paper, it is easier to understand the quality of the final policies. I believe the limitations of the approach as well as their justification are stated more explicitly this time. Furthermore, other issues regarding clarity, i.e., the superscripts of the latent variable model and why this approach is more computationally efficient than LAMBDA have been attended to. For these reasons, I have updated my score on empirical novelty to 3 and bumped up the overall score to 8.

---

### Official Review · Reviewer_sCDz · 2022-10-25

**Confidence:** 3
**Correctness:** 3
**Technical Novelty And Significance:** 3
**Empirical Novelty And Significance:** Not applicable
**Recommendation:** 6

**Clarity, Quality, Novelty And Reproducibility:**

The contribution of the paper appears to be novel and original. An anonymized GitHub repository is provided for the purposes of reproducibility. The paper is generally well written, but the notation is quite confusing, and can hardly be understood without referring to other papers. The POMDP model includes several kinds of latent variables z, distinguished by superscripts, which apparently do not mean powers, but something else. It doesn't help that the function p_\psi is overloaded in five different ways in Equation 2, so it is hard to tell which parameter does what. Similarly, in Equations 4 and 5, it is not at all clear what M is - probably not the model, because the parameters \psi appear to be defining the model. Is it perhaps a data set of samples? But then, D is used for that in Algorithm 1.

**Strength And Weaknesses:**

The paper attempts to solve one of the most difficult decision problems possible, and achieves favorable empirical results in comparison to existing algorithms. Moreover, it does it by learning a latent state model in the form of a POMDP, which makes the paper very suitable for this conference. Although the verification is in simulation and on relatively simple navigation problems, the dimensionality of the observation space is high, because observations are images, so solving this kind of problems is far from trivial, especially in the partially observable setting.

**Summary Of The Paper:**

The paper proposes an algorithm for solving safety-constrained decision problems with unknown dynamics that are also partially observable, which is one of the most difficult type of decision problems. The approach is to learn a safety-constrained POMDP model that has latent state, and also includes a safety critic. All components of the model are estimated simultaneously from experienced transitions, by modifying the training objective from only maximizing cumulative discounted reward to also maintain cumulative costs, in the sense of safety violations, under a specified threshold. Empirical evaluation in several environments shows computational advantages with respect to existing baseline algorithms.

**Summary Of The Review:**

This paper appears to be a good contribution to the field of learning and solving POMDP problems, with the further complication of safety constraints. The explanations of the algorithm and the mathematical notation can be improved.

---

> ### Author Response · Authors · 2022-11-11
> **Author Response**
>
> Dear reviewer sCDz,
>
> Thank you for the thoughtful review and for pointing out that this is not a trivial problem and ICLR would be a suitable venue for this work. In what follows, we would like to address and clarify  the concerns raised in the review.
>
> > latent variables z
>
>
> We decided to present a simplified version of the latent model in the main document, which contains a single latent variable. Since we found that the model we used is important for the overall performance of the Safe-SLAC algorithm, we also moved the detailed version to the supplemental material and added a more extensive explanation of the model.
>
>
> > p_\psi is overloaded in five different ways in Equation 2
>
> We changed the notation in Equations 2 and 3, including the superscripts $z,x,r,c$ in the parameters of the different function of the model. However, to avoid cluttering the notation, we mention that $\psi$ is used instead of $\psi^z, \psi^r,\psi^x,\psi^0$ thereafter.
>
> > it is not at all clear what M is
>
>
> Thank you for pointing this issue out. We added the following sentence to clarify that $M$ indeed refers to the model:
> "We use $M$ to refer to the functions of the model ($p_{\psi^z}$,$p_{\psi^x}$, $p_{\psi^r}$, $q_{\psi^0}$, $q_{\psi^z}$)"
>
> We hope this makes the notation clear. Please let us know if this addressed your concerns since we are happy to try and make any more appropriate changes, as needed.

---

> > ### Comment · Reviewer_sCDz · 2022-11-30
> > **Response to the authors' response**
> >
> > Thank you for addressing my comments on the notation used in the paper. I think the revised version has eliminated the possibility of confusion, and is easier to read and understand. Regarding the concerns of some of the other reviewers that little in this paper is original, the fact remains that the proposed Safe SLAC algorithms is still novel. Even if it might be obvious and straightforward to some readers how to extend the original SLAC algorithm for safety, for the majority of readers it would not be obvious, so the paper proposes a valid novel solution to a difficult decision problem that some practitioners would face. I think this justifies well the acceptance of the paper. For this reason, I will maintain my rating of 6.

---

### Official Review · Reviewer_kqqD · 2022-10-27

**Confidence:** 3
**Correctness:** 3
**Technical Novelty And Significance:** 1
**Empirical Novelty And Significance:** 1
**Recommendation:** 5

**Clarity, Quality, Novelty And Reproducibility:**



The method of the paper is very clear.

this paper is a combination of existing methods to me and is not novel.

The author also presents open-source code which is good for reproducibility.



**Details Of Ethics Concerns:**



**Strength And Weaknesses:**



Strength:

The presentation of the paper is very clear.

The experimental results are also sufficient to show the effectiveness of the proposed methods.



Weakness:

The main weakness is that the proposed method is not novel. The author claimed that 3 contributions of the method. They are minor to me. In my view, this paper extends SLAC to safe RL case by introducing the cost mechanism to the prediction target and discounted cost return, respectively. None of them are novel.



**Summary Of The Paper:**



This paper extends the stochastic latent actor-critic method to safe reinforcement learning by a safety critic. The experiments show the competitive performance of the proposed methods over existing approaches on benchmark pixel input tasks.

**Summary Of The Review:**



Generally, this paper is a combination of existing methods to me and is not novel. The paper takes a lot of content on introducing existing methods (constrained POMDP, SLAC). Using the Lagrange multiplier is also not novel for safe RL [1,2].





[1] Geibel, P. and Wysotzki, F. Risk-sensitive reinforcement learning applied to control under constraints. CoRR, abs/1109.2147, 2011. U

[2] Stooke A, Achiam J, Abbeel P. Responsive safety in reinforcement learning by pid lagrangian methods[C]//International Conference on Machine Learning. PMLR, 2020: 9133-9143.

---

> ### Author Response · Authors · 2022-11-11
> **Author Response**
>
> Dear reviewer kqqD,
>
> We appreciate the comments regarding **presentation** of the paper and the **effectiveness** of the Safe SLAC method. We are also glad to see our efforts to make the experiments **reproducible** being appreciated.
>
>
> > In my view, this paper extends SLAC to safe RL case by introducing the cost mechanism to the prediction target and discounted cost return, respectively.
>
> Regarding the **novelty concerns**, we agree a number of elements of our approach come from prior work. However, we would like to reiterate that such a combination has not yet been investigated and it is certainly the first time it is employed to constrained POMDPs with high dimensional observation spaces. **Reviewer 4rUn** summarizes this point: "The method mostly consists of previously established work that is combined in a new way. I see this as a strength."
>
> We would also like to mention another important point. The results we obtained can easily be reproduced. This could serve as a benchmark for future research in safe RL, as **reviewer asV6** mentioned.
>
> > The author claimed that 3 contributions of the method.
>
> We would like to respectfully clarify this point. The main contribution of our paper is indeed the introduction of the Safe SLAC algorithm, as the reviewer suggests. The three points in the introduction refer to details of how we "extend SLAC [...] to create our safe RL approach for partially observable settings (Safe SLAC)." To make that more clear we rephrased the sentence describing the contribution in the introduction:
> "We propose Safe SLAC, an extension of the Stochastic Latent Actor Critic approach (SLAC; Lee et al., 2020) to problems with safety constraints."
>
>
> > The paper takes a lot of content on introducing existing methods (constrained POMDP, SLAC)
>
> Thanks for pointing this out. Taking this comment into account and suggestions from other reviewers, we simplified the exposition of the SLAC model in the main document. Moreover, we moved the detailed description of the model to the supplemental material.
>
> > the Lagrange multiplier
>
> We changed our description of the SAC-Lagragian algorithm, first citing [1] and [2] as sources to the lagrangian relaxation approach for RL under constraints, and then mentioning that SAC-Lagragian is the application of this approach to the SAC algorithm.
>
>
>
> Finally, we would like to thank the reviewer for the constructive feedback. We hope to have addressed all the concerns raised. Please let us know if there are any further questions.

---

### Author Response · Authors · 2022-11-11
**General Response**


We would like to thank all the reviewers for their constructive comments. In summary, the reviewers largely agree the paper is well written and provides sufficient empirical evidence to demonstrate the effectiveness of the proposed method.

We made a number of changes to our paper in order to improve the presentation according to the specific comments of the reviewers, indicated in blue in the revised version. We believe the paper is more clear now. We list the main improvements below.

1. Simplification of the latent variable model in the main document (Equations 2 and 3). In particular, we present a latent variable model with a single latent variable $z$.
2. Included the full description of the factorization of the latent variable model in Appendix A.
3. Notation improvements, making clear how $\psi$ aggregate the parameters of different functions of the latent variable model.
4. Discuss some of the limitations of using a latent variable.
5. Include an unconstrained baseline to provide perspective on the quality of the final policies.
6. Swapped Figures 3 and 5 to remove the DoggoGoal outlier from the main document.
7. Added a qualitative analysis of the final policy rollouts based on the videos from the supplemental material (Appendix D).
8. Clarified the discussion on computational requirements.

---

### Decision · Program_Chairs · 2023-01-20

**Decision:**

Accept: poster

**Justification For Why Not Higher Score:**

* The approach is a combination of exiting techniques, but the combination and its application are novel
* The improvement with respect to previous techniques is okay but not significant

**Justification For Why Not Lower Score:**

* The paper proposes a method for safe RL in POMDPs that is competitive with the state-of-the-art.
* The revised version of the paper is clear
* This is good work that advances the state of the art in safe partially observable RL.   This work will be of interest to the RL community.


**Metareview: Summary, Strengths And Weaknesses:**

The paper describes a new technique for safe RL in the context of partially observable environments. This is done by using a latent variable with a safety critic.

Strengths:
* The paper proposes a method for safe RL in POMDPs that is competitive with the state-of-the-art.
* The revised version of the paper is clear

Weaknesses:
* The approach is a combination of exiting techniques, but the combination and its application are novel
* The improvement with respect to previous techniques is okay but not significant

Overall, this is good work that advances the state of the art in safe partially observable RL.   This work will be of interest to the RL community.

**Note From Pc:**

if the above contains the word "oral" or "spotlight" please see: "oral" presentation means -> notable-top-5% and "spotlight" means -> notable-top-25%. As stated in our emails, we are disassociating presentation type from AC recommendations